# African Swine Fever Virus (ASFV): Biology, Genomics and Genotypes Circulating in Sub-Saharan Africa

**DOI:** 10.3390/v13112285

**Published:** 2021-11-15

**Authors:** Emma P. Njau, Eunice M. Machuka, Sarah Cleaveland, Gabriel M. Shirima, Lughano J. Kusiluka, Edward A. Okoth, Roger Pelle

**Affiliations:** 1Biosciences Eastern and Central Africa—International Livestock Research Institute Hub, P.O. Box 30709, Nairobi 00100, Kenya; E.Machuka@ciar.org (E.M.M.); e.okoth@cgiar.org (E.A.O.); R.Pelle@cgiar.org (R.P.); 2Nelson Mandela African Institution of Science and Technology, Arusha P.O. Box 447, Tanzania; sarah.cleaveland@glasgow.ac.uk (S.C.); gabriel.shirima@nm-aist.ac.tz (G.M.S.); ljmkusiluka@mzumbe.ac.tz (L.J.K.); 3College of Veterinary Medicine and Biomedical Sciences, Sokoine University of Agriculture, Chuo Kikuu, Morogoro P.O. Box 3015, Tanzania; 4Institute of Biodiversity, Animal Health and Comparative Medicine, College of Medical, Veterinary and Life Sciences, University of Glasgow, Glasgow G12 8QQ, UK; 5Mzumbe University, Morogoro P.O. Box 1, Tanzania

**Keywords:** arbovirus, African swine fever, virus, swine, pigs, Africa, genotyping

## Abstract

African swine fever (ASF) is a highly infectious and fatal haemorrhagic disease of pigs that is caused by a complex DNA virus of the genus *Asfivirus* and *Asfarviridae* African suids family. The disease is among the most devastating pig diseases worldwide including Africa. Although the disease was first reported in the 19th century, it has continued to spread in Africa and other parts of the world. Globally, the rising demand for pork and concomitant increase in transboundary movements of pigs and pork products is likely to increase the risk of transmission and spread of ASF and pose a major challenge to the pig industry. Different genotypes of the ASF virus (ASFV) with varying virulence have been associated with different outbreaks in several countries in sub-Saharan Africa (SSA) and worldwide, and understanding genotype circulation will be important for ASF prevention and control strategies. ASFV genotypes unique to Africa have also been reported in SSA. This review briefly recounts the biology, genomics and genotyping of ASFV and provides an account of the different genotypes circulating in SSA. The review also highlights prevention, control and progress on vaccine development and identifies gaps in knowledge of ASFV genotype circulation in SSA that need to be addressed.

## 1. Introduction

African swine fever (ASF) was initially detected in Kenya a century ago as the cause of high fatality in domestic pigs, *Sus scrofa domesticus* [1]. The causative agent of the disease is ASF virus (ASFV), a double-stranded, icosahedral, enveloped arbovirus; DNA of the virus of the genus *Asfivirus* and *Asfarviridae* family [2]. The complete genome sequencing of this virus has identified approximately 160 genes which are organised in an almost similar manner to those of poxviruses. The incubation period of ASF varies from four days to about three weeks, depending on the virulence of the virus, host-related factors, the viral load and route of infection [3]. Clinical signs of ASF may resemble those of other haemorrhagic illnesses like salmonellosis, classical swine fever and erysipelas. The manifestation of ASF can be per acute, acute, subacute, and chronic forms. In the per acute form induced by the highly virulent ASFV strains, up to 100% pig mortality can occur four days after infection without gross lesions [4]. 

Apart from domestic pigs, bush pigs (*Potamochoerus porcus*), warthogs (*Phacochoerus africanus*) and ticks that parasitise them (*Ornithodoros moubata* spp.) have also been identified as carriers of the virus, making them possible reservoirs [5] that can promote the prolonged existence of the infection in an area for over five years [6]. For instance, the presence of ASFV in ticks for at least four years in unoccupied domestic pig premises in Madagascar was documented by [7]. The first documentation of ASF was in 1921 in Kenya and from there it has been reported in other countries inside and outside of Africa [8]. Reports of ASF outbreaks and the presence of ASFV in domestic pigs in approximately 32 African countries affirm its high endemic nature and continued threat to the swine industry in SSA [8].

A rapid polymerase chain reaction (PCR)-based sequencing of a 478 base pair fragment at the C-terminal end of the p72 capsid protein coding gene of the virus has commonly been employed to differentiate its different genotypes [9]. The value of genotype analyses and genomic epidemiology in general is in the finding that ASFV strains are currently grouped into 24 genotypes with all the gene variants being associated with the disease [10,11]. While most of these genotypes have been linked to ASF outbreaks in various parts of SSA [12], genotype I dominates in Central Africa [13,14] and West Africa [15,16]. Even though some genotypes of ASFV are country-specific in East and Southern Africa, others have a transboundary distribution [17]. For instance, most Tanzanian ASF outbreaks have been linked to genotypes II, IX, X, XV and XVI of the virus [18,19,20], while the highly virulent genotype II of the virus has been identified in Mauritius, Madagascar, Tanzania and Zimbabwe [20,21]. In addition, genotypes IX and X were encountered in Uganda and Kenya [22,23]. 

The last decade has seen a rapid increase in the pig production sector in Africa and a corresponding increase in detection of ASFV, which have altogether provided a better epidemiological picture in ASF-affected regions. Recently (June 2020), ASF cases reported from Africa are all from the SSA region as shown in Figure 1.

All of the known 24 different ASFV genotypes have been established in SSA and an updated literature on the genotypes circulating in SSA is still needed. According to [24], understanding the genetic and genomic variations of ASFV is crucial to understanding its diversity. Owing to the unavailability of treatment and vaccine for ASF and its complex epidemiology, knowledge about the genotypes circulating in SSA can improve its prevention and control strategies [25]. The present review briefly revisits the biology, genomics and genotyping of ASFV and provides an account of the different genotypes circulating in SSA. Information assembled in this work may help identify the predominant genotypes in SSA and promote ASF research and control in the region.

## 2. Methodology

A literature review was carried out to uncover information on African swine fever in sub-Saharan Africa. Several publications written in English language on ASF were retrieved mainly from Google Scholar, HINARI and Research for Life databases by keywords searches. Downloaded publications were saved in a Zotero v 5.0.93 library, which was also used as a reference manager. Screening was done by reading the papers to identify those from accredited sources with a preference to open access articles published under a Creative Commons Attribution 4.0 (CC-BY 4.0) license that permits users to copy, modify, or distribute data in any format for any purpose, including commercial use. Initially, a detailed literature search was directed towards identifying the review papers and articles in the field that pinpoint gaps and priorities for ASF research. Such information was used as a guideline to determine where to focus the current review. Paper sub-headings were decided upon and classified based on the ease of information flow. A paper map composed of subsection headings was drafted, forming a skeleton for the review. Articles were grouped based on their association with keywords relating to the draft sub-headings. Additional information on pigs and pig diseases was retrieved from The Pig Site, World Organisation for Animal Health (OIE), World Animal Health Information system (WAHIS), Food and Agriculture Organisation (FAO) and Global African Swine Fever Research Alliance (GARA) webpages. The National Centre for Biotechnology Information (NCBI), African Swine Fever Virus Database (ASFVdb) and European Nucleotide Archive (ENA) were the primary sources of information on publicly available ASFV genome sequences.

## 3. The African Swine Fever Virus Biology

The ASFV is a large DNA virus with multiple envelopes and the only virus classified under the *Asfarviridae* family and *Asfivirus* genus [26]. The virus encodes 150–165 proteins with essential functions in replication and evasion of host immune responses [2]. A proteomic atlas of the ASFV has been provided by [27]. Although about 54 of these proteins are structural [2], both structural and infection-related proteins have a role in modulating or regulating the host immune system evasion mechanisms, such as inhibiting host transcription factors that affect its replication [28]. ASFV entry into the cell is through the host’s cell surface receptors (Figure 2) following an infection of the animal mostly via the tonsils closest to lymph nodes.

Following viremia, the virus migrates to the tissue organs. Direct contact with infected pigs, swills or contaminated fomites forms the main route of exposure [3]. *Ornithodoros* ticks become infected with the virus when feeding on infected pigs; the virus replicates in their guts and migrates to salivary glands. Ticks infected with ASFV can bite pigs and directly introduce the virus into the blood, acting as vectors for the virus from one animal to another. Entry of the virus into pig cells is by endocytosis which can be clathrin or receptor-mediated [30,31].

When affecting a new herd, ASF is manifested by massive deaths of animals following high fever, loss of appetite and reduced movement, pigs clumping together, and in the severe form of a disease, death may occur four days before other clinical signs are observed. Appearance of clinical signs of the disease may be dependent on viral genotype, animal breed, environmental conditions, incubation period and route of exposure [32].

Petechial haemorrhages, mucoid diarrhoea and reddening of the skin around the ears, stomach and limbs are common in the less severe form of the disease (Figure 3) [33,34]. When a post-mortem is performed, haemorrhages are seen in internal organs, including the lungs, lymph nodes, intestines, heart, kidney and liver. The spleen increases in size and becomes darker than normal [35]. The role of scavenging pigs, free-living hosts and pigs recovering from ASF in the epidemiology of the disease is still questionable, although some studies have mentioned that they play a role in virus transmission [36,37].

## 4. African Swine Fever Virus Characterisation, Genomics and Genotyping 

Multiple methods have been used to characterize the ASF virus [38]. Biological markers and properties displayed by the virus in cultures are among the commonly used approaches. When cultured in the laboratory, cytopathic effects of infected cells and hemadsorption of red blood cells have been used to characterise the virus into either hemadsorbing or non-hemadsorbing strains [31]. Infectivity and cell response caused by the virus on cell cultures have also been used to characterise the virus as being high, moderate or low pathogenic [39]. Structural proteins, digestion by restriction endonuclease enzymes and the number of multigene families are also among different ways that have been used to study the virus [40,41,42]. 

With advancements in techniques, including partial and whole genome sequencing, molecular techniques have emerged as the major approach to characterising the viruses. The ASFV genome comprises a conserved central region of approximately 125 kb and two-variable ends that code for five multigene families (MGFs). The presence or absence of a MGF explain for the variable genome size of ASFV ranging from 170 to 193 kb [2]. Numerous MGFs are instrumental in determining the virulence of ASFV and the deletion of particular MGFs has resulted in attenuated phenotypes that facilitate immunity against challenges to its virulence [43]. Although the critical role of MGFs in ASFV virulence has been demonstrated, the ability of the virus to develop antigenic variability and escape the host immune response is not clear [26]. Likewise, the relationship of some genes in MGFs to host protection has not been fully elucidated.

Currently, four different gene regions, p72, p30, p54 and B602L, are commonly used to identify ASFV genotypes [23,44,45,46,47]. The genotypic differentiation of ASFV genotypes is largely dependent on the amplification and sequencing of the variable 3′-end of the B646L gene that encodes p72 which is the main capsid protein [48]. The sequencing of the B646L gene has defined 24 genotypes of the virus (I–XXIV) [10,14,26]. Most of these genotypes can be geographically mapped to locations, for example, genotype I is predominant in West and Central Africa, while genotype II has been mostly detected in eastern and southern parts of SSA [49]. The recent detection of the novel genotype XXIII in Ethiopia [10] suggests that there could be other ASFV genotypes yet to be documented in SSA [26]. There is also a possibility that the currently known 24 genotypes may be widely distributed than currently known censuring the low surveillance rate in new areas in SSA. Analysis of the tandem repeats in the central variable region of the B602L gene [46] or the intergenic region between the I73R and I329L genes at the right end of the genome [50] can be used to distinguish closely related ASFV isolates. The B602L gene is a particularly discriminative genetic marker whose sequencing has distinguished up to 31 subgroups of viruses with varying tetrameric amino acid repeats [47]. Many other gene regions, such. as the E183L, CP204L and EP402R encoding the p54, p30 and CD2v proteins, respectively, have also proved valuable in the analysis of ASFV from various locations to trace its spread [23,46]. 

### Molecular Epidemiology of African Swine Fever Virus in Sub-Saharan Africa

Extensive molecular epidemiological studies of the ASFV genotypes circulating in Africa have shown a great diversity of ASFV isolates in the continent [47]. Some studies that have reported ASFV genotypes circulating in SSA are listed in Table 1. The complexity of the ASF epidemiology has demonstrated the existence of all of the known 24 ASFV genotypes in SSA based on the sequencing of the p72 protein gene [9,48]. 

The infection of susceptible animals can occur either through the sylvatic or the domestic cycles [57]. In the sylvatic cycle, the ASFV circulates among wild reservoirs like bush pigs, warthogs and soft ticks (*Ornithodoros* spp.) [17]. On the other hand, the domestic cycle occurs when the virus is transmitted among domestic pigs independent of sylvatic hosts or arthropod vectors [17]. The transmission of ASFV from sylvatic to domestic hosts is assumed to occur when there is a spill over due to infected ticks as well as feeding or scavenging contaminated swill by domestic pigs at the wildlife-livestock interface [11,79]. This warthog-tick sylvatic cycle is linked to the diversity of the ASFV circulating in certain areas in SSA, particularly where the wildlife–livestock interface occurs [8,80]. On the other hand, the continued spread of the virus in domestic pigs may be attributed to virus spill from wild boars or the properly established domestic transmission routes that involve the movement of infected pigs or pig products or fomites [3].

The sylvatic cycle of the ASF in several countries in eastern and southern Africa has been described in detail [57]. While most of the ASFV genotyping studies in SSA have dwelled on the domestic cycle of transmission (Table 1), genotyping of ASFV in African warthogs has largely been overlooked, yet they are considered as the main wild vertebrate hosts of the ASFV where the disease is endemic [1]. For instance, most surveillance studies on the circulating genotypes of ASFV in Tanzania have been limited to outbreaks in pig farms [19,42,70]. Although other studies have tried to identify ASFV genotypes in African warthogs and ticks [23,44,66,68,74,81], no conclusive results have yet been obtained. For example, although seropositive ASFV samples was established in all sampled warthogs in the Serengeti ecosystem in Tanzania [81], the genetic makeup of the ASFV strains was not confirmed. Similarly, studies at the wildlife–livestock interface of the Gorongosa National Park in Mozambique generated only serological rather than genetic evidence of infection in warthogs [11]. Another study carried out in Saadani National Park in Tanzania revealed the presence of genotype XV in ticks collected from warthog burrows even though the genotype responsible for ASFV antibodies in warthogs could not be established [68]. For the remaining parts of SSA, consistent data regarding the occurrence of sylvatic cycles is lacking. 

Although warthogs are extensively distributed across the savannahs of West Africa and northern parts of Central Africa, the classic sylvatic cycle of ASFV has not been demonstrated in these parts of SSA, possibly due to the lack of comprehensive surveillance and partly because of the absence of *O. moubata* ticks in these regions [82]. As such, the transmission of ASFV in West Africa is thought to occur in domestic pigs independently of sylvatic hosts [16], and most outbreaks are associated with the movement of infected pigs or swine products [23]. It has been suggested that these contrasting epidemiological transmission patterns could be attributed to the greater genetic variability of ASFV isolates from eastern and southern Africa that comprise 22 distinct genotypes as opposed to the high homogeneity in West African ASFV isolates of genotype I [18,48]. 

The rich genetic diversity of ASFV is not only promoted by the sylvatic cycle alone but is also enhanced by the domestic cycle through the open borders and uncontrolled movement of pigs and swine products. For instance, the transboundary spread of ASF in eastern Africa during outbreaks is mainly associated with the horizontal transfer of ASFV between pigs due to the unregulated movement of pigs and pig products across borders [19,20,21]. Several studies have analysed the role played by cross-border pig movements concerning the outbreak of ASF in countries such as Kenya, Mozambique, South Africa, Tanzania, Uganda and Zambia [18,19,22,23,46,48]. For instance, during the 2010 ASF outbreak in Tanzania, genotype II of the virus was presumed to be introduced into Tanzania from Malawi through cross border transmission [20]. Similarly, the possibility of transboundary transfer of ASF in East Africa has been emphasised, particularly between Uganda and Kenya [23,46]. ASF outbreaks in Kenya between 2006 and 2007 were due to the fact of a genetically similar virus to that isolated in Uganda during the same time [23]. Although the western and central parts of Africa have normally detected the existence of genotype I of the virus with low genetic variability, the dissemination of ASFV genotypes to western from eastern Africa has also been established [18,23,48]. Even though more explanation might be needed regarding the evolutionary drivers of genetic diversity and probably different mechanisms might be proposed, it is evidenced from the above studies that the ASFV genotypes initially believed to be specific to certain geographical locations have now spread wider.

The complex epidemiological pattern of the ASFV is evident in many countries of SSA where outbreaks have been reported. The genetic characterization of ASFV using standardized genotyping procedures during the 2013–2015 outbreak in Zambia linked the outbreaks to ASFV genotypes I, II and XIV [76]. In the study in [71] on the genotyping of five ASFV isolates from domesticated swine in Uganda, four sequences were found to be similar to one another and closely linked to the only established genome sequence of p72 genotype IX. In Kenya and Uganda, two ASFV genotypes, IX and X, have been observed to be dominant in domestic pigs, warthogs and ticks [23,46,57]. Genotype I of the ASFV, which is considered to be highly fatal was linked to the random but persistent outbreaks of ASF in both confined and unconfined small-scale Kenyan swine farms between 2005 and 2011 [23,46,60]. According to [11], Malawi seemingly has the highest number of genotypes VIII ASFV variants, many of which are common in its neighbours, Mozambique and Zambia, indicating the possibility of Malawi being a possible reservoir of the infection and the need for regional collaboration for successful containment of the disease. Some novel ASFV genotypes previously not established in SSA have also been identified, for instance, genotype XXIV in Mozambique [11] and XXIII in Ethiopia [10]. Nevertheless, the most complex and diverse blend of ASFV genotypes has been reported in the eastern and southern region of Africa, as seen in Figure 1, where the transmission of the virus occurs via both sylvatic and domestic cycles [11]. 

There is mounting evidence of the ability of apparently healthy pigs or persistently infected animals to act as carriers of ASFV. Such carriers are likely to contribute to the maintenance of ASFV in pig production systems where clinical ASF outbreaks occur repeatedly. Studies in Uganda [22] and Kenya [56,57,60] have established ASFV sequences in healthy pigs, suggesting that less virulent strains could be circulating in these countries. A separate study in [53] established the presence of ASFV in apparently healthy pigs in parts of the DRC. The disease is also increasingly becoming endemic in parts of south-eastern Africa in domesticated pigs that carry the virus but have gained resistance to it [83]. Most investigations into the existence of ASFV in asymptomatic pigs have purely been surveillance studies and the genotypes of ASFV were not established. It is not known how the virus persists in endemic pig populations, although it is assumed that survivors, sub-clinical and chronically-ill pigs all contribute to the maintenance of the virus [4]. Genotyping also suggests that the virus isolates associated with this condition can be of low or high virulence [56]. The existence of long term carriers of ASFV has been disputed by some researchers [84] even though some of the cited studies provide evidence that isolates with reduced virulence could be circulating in SSA.

## 5. ASF Prevention, Control and Vaccine Development

African swine fever is a transboundary animal disease [85,86,87] and the focus for prevention and control includes both formal and informal interventions in national and international trade [88] Even though quarantine measures are known, disease introduction pathways are difficult to understand and use for risk estimation [89]. Moreover, although they are often put in place, biosecurity breaches have been often observed and reported [90,91]. Evaluation of risk factors associated with occurrence or re-occurrence of the disease in an area has been used to provide information, for instance in returning trucks and wastes from international ships and planes, although such information has not been practical enough to prevent the occurrence of the disease [92,93]. As has been the case with most animal diseases involving wild and domesticated animals, methods such as vector control and challenges from wildlife reservoirs may require some modifications in the natural environment, especially in the wild, rendering them difficult to implement [94]. Mathematical models that simulate control options for ASF based on their status involving susceptibility, persistently infected versus asymptomatic animals and deaths induced by the disease have been established to help predict optimal timing that is appropriate for a specific control intervention such as the use of biosecurity measures [95]. 

Most of the time, confirmation by detection of ASFV is done in OIE-approved laboratories [96]. The approved laboratories provide remarkable technical capacity and acceptable biocontainment. These laboratories are often located several kilometres away from the outbreak areas that require shipment of samples over long distances, often using ground transport and thus delaying diagnosis. Selected control measures such as movement restrictions and quarantine of infected properties can and are being implemented on suspicion of disease. While disease suspicion has been implemented for a rescue, it is however notable that delays associated with remoteness hinder outbreak responses especially for diseases, such as ASF, which have symptoms that are closely related to other diseases. Such a situation has been observed not only in animals but also in human viral diseases [97,98].

With good diagnostics and an efficient vaccine, it has been possible to control several endemic livestock diseases as shown by successes in control of FMD in south America, brucellosis in eastern Europe and blue tongue virus in Europe. There are constraints to the livestock and veterinary sector in SSA relating to both the lack of a commercial vaccine and the lack of infrastructure for delivery. This means that few livestock vaccination programmes have been implemented, and there remains a high risk of diseases spreading beyond the continent. For ASF, a further major constraint relates to the lack of commercially available vaccines. As part of ongoing efforts to develop vaccines against ASF, live attenuated ASF viruses have been used in vaccination experiments under controlled conditions with mixed results. Some approaches involved include adaptation of virus strains in cell culture to attenuate them, although, in these studies, such attenuated strains could not confer protection to the pigs upon exposure to virulent strains [99,100,101]. Deletion of genes associated with virulence has also been tried with some encouraging results as some trials showed up to 100% protection against African swine fever [28,102,103,104,105,106,107]. However, the attenuated strains could regain the lost virulence gene when tested in the natural host, or the dose difference between protective and lethal doses was too small and unsafe for the animal [105,108]. Therefore, because these live vaccines can cause chronic or persistent infections and can revert to virulence or recombine with field strains, an efficient attenuated vaccine against ASF is yet to be developed. In some recent studies, there is an observation of molecular divergence between African and European ASFV strains. The viruses recently derived from the latter suggest that live attenuated vaccines derived from Europe may not always cross-protect against ASFV genotypes derived from different regions of Africa, as it has been observed for phylogenetically distant ASFV strains within the p72 genotypes [109]. Techniques that have worked against other diseases, such as di-codon de-optimization in which the virus is kept alive while reducing its virulence, are still under trial [110].

Concomitantly, several studies have also been undertaken to develop subunit vaccines for ASF viruses, including an intensive search for good candidate vaccine antigens for ASF. This new approach would offer a safer vaccine without drawbacks and safety risks associated with live vaccines. Results obtained so far are promising even though they are still provisional [111,112,113]. However, increased infectivity, sometimes with enhanced clinical symptoms upon challenge with ASF, was reported in some experiments [114,115,116]. Nonetheless, through all these studies, significant progress has been made.

## 6. Emerging Gaps and Future Research Focus

Although some research has been conducted on ASFV genotypes in SSA, more research is still necessary to discover new ASFV genetic markers and those connected to the evolution of ASFV isolates, particularly in parts of SSA where they are endemic. Knowledge of novel genetic markers involved in virulence can be valuable in the control strategies against the virus, for example, it may assist in identifying high-risk routes of cross-border spread. In this regard, the genetic characterisation of MGFs virulence genes to classify ASFV isolates based on their virulence factors can be a practical tool for control of ASF [26]. A current study that released the first genotype II ASFV full sequence of an African origin highlighted that other parts of the genome other than the commonly used regions may be analysed to give improved resolution 

The existence of domestic pigs asymptomatic to ASF has largely been attributed to the evolution of ASFV towards moderate virulent forms [26]. However, the molecular factors determining whether ASFV infection becomes asymptomatic in African suids remain largely unknown and need more research [26]. Asymptomatic ASF is mostly reported in endemic cases. Pigs in endemic settings should be considered significant for propagating the variations in the virulence of isolates in circulation besides evaluating their roles in disease transmission and persistence. Experiments involving ASFV isolates from recovered animals will possibly enable an improved understanding of their transmission routes, their presence and continued existence in diseased tissues and a better characterisation of carriers on potential clinical activation [26]. There is also little empirical evidence for the transmission of ASFV from continually infected animals to naive ones leaving the importance of carrier animals in the field unclear.

Diagnosis and early detection of severe or rapidly spreading outbreaks of the virus on farms and fields could be improved by identifying the virulence factors and mechanisms of pathogenesis of ASF. In this regard, genomic markers associated with ASFV virulence should be established and fully characterised to provide for the designing of improved and more suitable diagnostic approaches based on the expected clinical signs in infected animals [26].

Although the classic sylvatic cycle has not been established or adequately investigated in many countries, it is evident that this cycle exists in several SSA countries [12,117]. Of particular interest are the borders of protected areas in eastern and southern Africa, where the cohabitation of warthogs and *Onithodoros moubata* ticks is likely to be a principal factor in propagating ASFV from wild hosts into domestic pigs [118]. Although wild ASFV carriers like bush pigs (*Phacochoerus larvatus*) and giant forest-hogs (*Hylochoerus meinertzhageni*) exist, their part in ASF maintenance and transmission remains unclear. As much as ASFV is maintained in asymptomatic wild suids, such as warthogs (*Phacochoerus africanus*), in Africa, most surveillance studies have been restricted to pig farms [19,42,70]. Some studies have reported the occurrence of a sylvatic cycle independent of domestic pig involvement [68], but more research focusing on this area is needed for a solid conclusion. Furthermore, although both cycles have been described in eastern and southern Africa [9,10,48], the sylvatic cycle has not been elucidated in western Africa [23] and more research is required in this regard. 

The complexity of ASF epidemiology is evident in East Africa, where the characterisation of the p72 gene and immuno-dominant protein VP72 have established the presence of discrete genotypes [18]. Nevertheless, the low levels of intra-genotypic variation (0.1%) across this gene region continue to preclude determining the origin, subsequent spread or relatedness of these outbreaks. Contemporary studies suggest the likelihood of more ASFV genotypes in Africa [10], and the potential of other genetic markers could prove useful in intra-genotype classification [50]. 

The gaps outlined in the current review narrow down to an emphasis on the importance of continuous surveillance and generating more information on ASFV significantly through whole genome sequencing. Recent developments in whole-genome sequencing can enable complete genotyping and provide essential data needed to elucidate the ASF biology and diversity in SSA. While the quantity of ASFV genomes available in public databases continues to increase, additional genome sequences of distinct and historical ASFV isolates are necessary [24]. Unfortunately, the genotyping of p72 does not always enable sufficient typing resolution or discrimination between different biological phenotypes of the virus [73]. Though not exhaustive, an increased genotyping resolution can be attained by further assessments of p54, p30 and B602L genes [18,44,46,47]. Fully sequenced genomes can be used to extract more information. For example, when the genotype II virus from Tanzania was compared with a genotype II virus from Georgia, the p72, p54 and p30 regions indicated that the virus strains were identical. Upon a comparison of their full genomes, it was observed that the two virus strains had a series of divergences in 5′ and 3′ multicopy gene families (MGFs). Apart from that, there was also an insertion of 16 bases that yielded a tandem repeat (AAAAAAATAAACAACA) located at the 5ʹ of the ATG codon of locus KP177R that encodes for the K177R structural protein. Analysis of the numbers of reads confirmed the presence of this mutation which differentiated Tanzania/Rukwa/2017/1 from all currently available genotype II genome sequences. In addition to the multiple changes in the terminal multicopy gene families, the single-copy gene appeared to be absent from the Tanzanian strain but it was present in the Georgia 2007 strain and its derivatives. This is a practical example of what improved resolution can contribute on practical control of ASFV and in the understanding of its epidemiology and evolution. Unfortunately, despite the long history of research on ASFV and the overwhelming potential of transboundary transmission, the volume of publicly available ASFV genome sequences is still inadequate. Currently, the GenBank database contains approximately 54 complete ASFV genome sequences. More complete genome sequences of ASFV are needed to comprehend the plasticity, antigenic diversity and evolution of its genome [24]. 

## 7. Conclusions and Recommendations 

Even though ASF was first detected in Africa a century ago, many gaps remain in our understanding of the epidemiology of ASFV and the genotypes circulating in SSA, with few complete sequences directly originating from this region. The major outbreaks of ASFV in Asia demonstrate the ability of the virus to spread beyond Africa, with devastating consequences, highlighting the importance of a better understanding of the epidemiology and transmission biology of the virus to prevent spread within and from endemic regions of Africa. Moreover, there are still numerous challenges in our understanding of the biology of the virus. Studies have already shown that some of the ASFV strains circulating in SSA are highly virulent and can kill most infected animals. As such, there is a need for constant monitoring, alongside the identification of ASFV genotypes with altered virulence. As a known ASF hotspot and almost considered home to globally circulating genotypes such as the pandemic genotype II strain, it is surprising that from over 20 publicly available complete genomes, there is only one strain of this genotype whose origin is directly linked to sub-Saharan Africa [69]. The increased usage of complete genome sequences of the virus may reveal more genetic and virulence markers from other genomic regions that can be used to track its spread than the less than ten currently used regions. Studies of this nature will eventually allow documentation of more ASFV genotypes and a better understanding of ASF epidemiology in SSA and beyond.

Moreover, molecular characterisation of the isolates circulating in the wider African region is necessary to establish and comprehend the evolution of existing isolates, particularly in SSA regions known to have persistent outbreaks as well as high genotype diversity. Based on this, the current study recommends future ASF and ASFV research to focus on surveillance of ASFV on new areas and availability of complete genome sequences of different genotypes from sub-Saharan Africa. Prioritisation should be given to field strains, and where possible, strains that have been associated with both sylvatic and domestic cycles should be of particular interest due to their wider implications in control strategies.

## Figures and Tables

**Figure 1 viruses-13-02285-f001:**
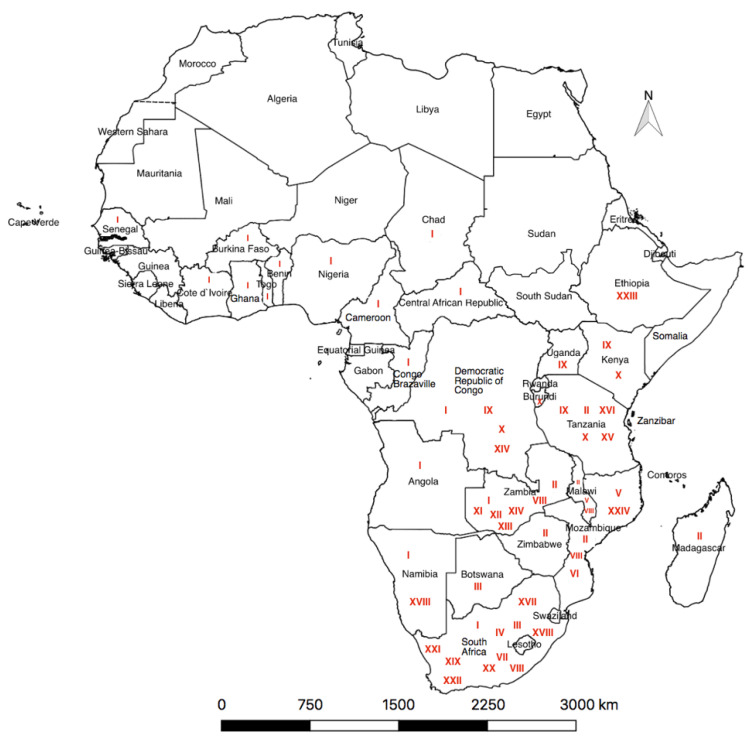
A continental distribution of ASF cases from 2016 to 2020 as reported to OIE. Active cases were reported from 32 African countries all falling under the SSA region. The richest diversity of ASFV p72 genotypes (written in red) is observed in the eastern and southern parts of Africa compared to the western, central and northern parts.

**Figure 2 viruses-13-02285-f002:**
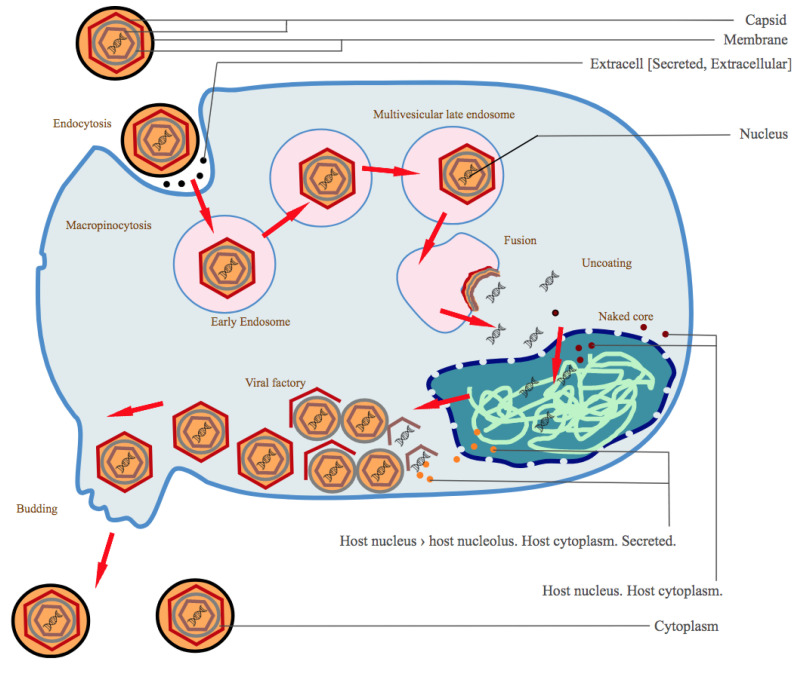
Mechanism of ASFV entry into the animal cell, replication and release. Image from [29].

**Figure 3 viruses-13-02285-f003:**
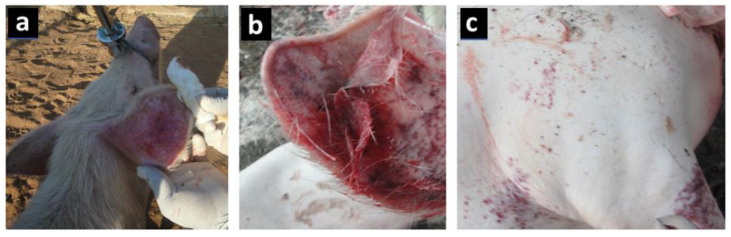
Images showing the clinical signs of ASF: (**a**) an animal with fever and turning red, especially in the ears and nose; (**b**) pinpoint haemorrhages observed in ear lobes; (**c**) petechial haemorrhages on the skin surface.

**Table 1 viruses-13-02285-t001:** Examples of ASFV genotypes that have been reported in sub-Saharan Africa.

Country	Genotype	Sample Source(s)	Year of Study/ Collection	Reference(s)
Burundi	X	Domestic pigs	2018	[16]
Cameroon	I	Domestic pigs	2010–2018	[51]
Cameroon	I	Domestic pigs	2018	[52]
Democratic Republic of Congo (DRC)	IX	Domestic pigs	2009	[23]
DRC	X	Domestic pigs	2016	[53]
DRC	IX	Domestic pigs	2011	[54]
DRC	I, IX, XIV	Domestic pigs	2005–2012	[25]
Ethiopia	XXIII	Domestic pigs	2011	[10]
Ivory Coast	I	Domestic pigs	2014	[55]
Ivory Coast	I	Domestic pigs	2008–2013	[13]
Kenya	IX	Domestic pigs	2006–2007	[56]
Kenya	IX	Domestic pigs	2007	[46]
Kenya	X, IX	Domestic pigs, soft ticks and warthogs	2005	[23]
Kenya	IX, X	Domestic pigs	2008–2009	[57]
Kenya	IX, X	Soft tick, Domestic pigs	2006	[58]
Kenya, Uganda	IX	Domestic pigs	2011–2013	[59]
Kenya-Uganda	IX	Domestic pigs	Not mentioned	[60]
Madagascar	Not mentioned	Soft ticks	2007–2008	[61]
Madagascar, West Africa, and Mozambique	I, VIII	Domestic pigs	2000	[9]
Malawi	I, II	Domestic pigs	2019	[44]
Malawi	V, VIII	Warthog, Domestic pigs	1960	
Malawi, Mozambique, Zambia	VIII, V	Domestic pigs	2001–2003	[18]
Mauritius	II	Domestic pigs	2007–2008	[49]
Mozambique	II, V, XXIV	Soft ticks	2007	[11]
Mozambique	II, VIII, V, VI	Domestic pigs	1998	[44]
Namibia	I and XVIII	Domestic pigs	2018	[62]
Namibia	Not mentioned	Domestic pigs	2018	[63]
Nigeria	I	Domestic pigs	2007–2015	[14]
South Africa	III, XX	Soft ticks	1985–1987	[64]
South Africa	III, XIX, XX, XXI	Soft ticks	1987–1996	[65]
South Africa	I, III, IV, VII, VIII, XIX, XX, XXI and XXII	Soft ticks, Warthogs	1987–2003	[66]
Swaziland, South Africa	XIX, VII, XVII–XXII	Soft ticks	1973–1999	[48]
Tanzania	II, IX, X	Domestic pigs	2015–2017	[67]
Tanzania	XV	Soft ticks	2017	[68]
Tanzania	II, IX	Domestic pigs	2015, 2017	[69,70]
Tanzania	X	Domestic pigs	2013	[42]
Tanzania	XV	Domestic pigs	2008	[19]
Tanzania	X	Domestic pig	2009	[20]
Tanzania	XVI	Domestic pigs	2001	[21]
Uganda	IX	Domestic pigs	2007	[23]
Uganda	IX	Domestic Pigs	2015	[71]
Uganda	IX	Domestic pigs	2015	[72]
Uganda	IX	Domestic pigs	2010–2013	[22]
Zaire (DRC)	I	Domestic pigs	1974–1989	[73]
Zaire, South Africa	IV, XX	Domestic pigs, warthogs	1977—Zaire 1999—S. Africa	[74]
Zambia	I	Domestic pigs	2015	[75]
Zambia	I, II, XI, XII, XIII, XIV	Domestic pigs	2013–2015	[76]
Zambia	II	Domestic pigs	2017	[77]
Zambia	VIII	Domestic pig	1988	[44]
Zambia, South Africa	I, III, XXII	Soft ticks	1983—Zambia 2008—S. Africa	[74]
Zimbabwe	II	Domestic pigs	2015	[78]

## Data Availability

No original primary data is associated with this review.

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
