# Peer review of "African Swine Fever Virus (ASFV): Biology, Genomics and Genotypes Circulating in Sub-Saharan Africa"

_viruses, 2021, doi:10.3390/v13112285_

Round 1
Reviewer 1 Report
This review article focuses on African Swine Fever Virus (ASFV): Biology, Genomics and Genotypes Circulating in Sub-Saharan Africa. In addition authors describe SF prevention and vaccine development and emerging gaps and future research focus extensively.
There have been a number of review articles published recently on ASF biology, vaccine development, control measures and also on genotypes and epidemiological findings in Sub-Saharan Africa. This review as it stands now, does not provide additional information to the scientific community. I recommend to remove the sections on ASFV biology and prevention and vaccine development, and focus on latest developments on ASFV genotyping and sub-genotyping work done with African isolates.
Minor comments:
There are many typos throughout the manuscript.
Table 1 is not comprehensive. Information on outbreaks reported in Angola, Benin, Botswana are not listed but described in the text for example.
Figure 1, 2 and Plate 2 is not necessary. Are the pictures in the Plate 1 and 2 were generated by the authors? Plate 2 C normal spleen does not appear like to normal pig spleen.
Author Response
Dear reviewer 1,
Thank you for your contribution to improve our manuscript.
The questions and responses have been addressed as follows:
Qn:There are many typos throughout the manuscript.
Resp: The manuscript has been re-read and revised.
Qn: Table 1 is not comprehensive. Information on outbreaks reported in Angola, Benin, Botswana are not listed but described in the text for example.
Resp: The Authors agree that Angola, Benin and Botswana have been described in the text but not listed in the table. The publications on ASF and ASFV from the three mentioned countries were difficult to retrieve as they did not meet the criteria used in our methodology. Reliable information on these countries was obtained from OIE but the sample sources, year of study and other published references to support such information could not be retrieved for inclusion in the table.
Qn: Figure 1, 2 and Plate 2 is not necessary. Are the pictures in the Plate 1 and 2 were generated by the authors? Plate 2 C normal spleen does not appear like to normal pig spleen.
Resp: Yes, the pictures in Plates 1 and 2 were generated by the authors
The authors agree that the information in figure 1 is represented in figure 3 and that in Plate 1 can be used to explain Plate 2 and changes ave been made. However, Figure 2 is still considered important for ASFV biology. Does the reviewer suggest a replacement of this figure with another figure or a complete omission without replacement?
Thank you.
Reviewer 2 Report
The review "African swine fever virus (ASFV): Biology, genomics and genotypes circulating in sub-Saharan Africa" provides a thorough literature review on the current status and distribution of ASFV genotypes in Africa. Many aspects of the viruses epidemiology are reviewed and reports of particular genotypes in different regions. The authors highlight the fact that there is no complete ASFV genome for the widespread genotype II that has been isolated from SSA and conclude that more extensive sequencing of the virus would improve genotyping and contribute to a greater understanding of the virus. However, the authors need to improve the argument for this. It is not clear from this review why complete genome sequencing (at >170 kilobases, not a trivial task) and an improved understanding of ASFV evolution (not a subject discussed in the review) will reveal more information about ASFV epidemiology and control. Why is intra-genotyping needed? Are some genotypes less virulent or attenuated than other? If so which?
The following comments should be addressed by the authors:
1) Why are certain words underlined in the Introduction?
2) Clarify the statement "Although the disease was first reported in 1921 in Kenya, it has now spread to most SSA countries...". Has it spread or just been detected?
3) revised the sentence on line 86 e.g. "The last decade has seen a rapid increase in the pig production sector in Africa and a corresponding increase in detection of ASFV...".
4) What does the phrase on line 72 "..many ASFV genotypes have been established in SSA...". The review has not considered the evolution of ASFV. The sylvatic cycle is only mentioned on page 7 and should be included in the Introduction. The existence of a wild swine-tick cycle in SSA is critically important to both the evolution of ASFV and its emergence in domestic pigs.
5) Why is ASFV a 'large' DNA virus?
6) The sentence on line 112 needs revision. "Entry of the virus into pig cells (Fig2) is mostly via the tonsils closest to the lymph nodes" does not make sense. Infection by fomites could infect the animal via the tonsils but virus infects cells through a cell-surface receptor, implied in Figure 2.
7) Line 115, virus in the bloodstream is normally referred to as viraemia. On line 119 ".. from one animal to another."
8) on line 147, explain what pathogenicity is in the context of cell culture?
9) When discussing the genome of ASFV, the authors should provide more background on the genome structure, nomenclature of ASFV genes and give more information on MGFs if they influence virulence.
The authors also do not consider the mutation rate of ASFV and whether this varies across the genome and how it has given rise to variation in key genes. This may be the only argument for complete genome sequencing.
10) The paragraph starting line 189 does not explain the the transmission cycle, which is admittedly complex (warthog-tick-warthog; warthog-tick-domestic pig; warthog-domestic pig; domestic pig-domestic pig), particularly well. The natural cycle of wild pig - tick should be described and then the interaction between this and the introduction of domestic pigs. The distribution of key species such as warthogs in Africa is not considered but may contribute to the distribution of ASFV is SSA.
11) Line 245: revise to "..on the genotyping of five ASFV isolates (or samples) from domestic..". Line 259. ".. where the transmission of the virus occurs.."
12) The discussion on carrier status is mentioned on line 263 and again on 346. As this is a controversial subject, it only need one mention.
13) On line 299, the discussion suggests that a delay in diagnosis is a major issue. Whilst accurate diagnosis of disease is important, some control measures can be implemented (movement restrictions / quarantine of infected properties) on suspicion of disease and can be effective, so whilst inconvenient, it is not an insurmountable problem.
14) Line 392, why is genotyping using p72 not always sufficient for typing resolution or phenotype discrimination? What did further assessment using p54, p30 and B602L genes provide? Despite 12 pages of text it is not clear what the improved resolution gives for practical control of ASFV or in the understanding of its epidemiology/evolution. Based on the review it is surely possible to give a more nuanced recommendation than more sequencing across SSA. Should effort be directed at areas with persistent outbreaks or regions showing high genotype diversity.
Author Response
Dear Reviewer 2,
We thank you for your contributions towards the improvement of our manuscript.
We have done the changes which can be tracked in our improved manuscript. The point by point responses to the questions raised a re as follows:
Qn: Why are certain words underlined in the Introduction?
Resp: Words underlined in the introduction are the key words and search terms that have been automatically identified by the journal database. Clicking on the underlined words will link the reader to other information related to the words.
Qn: Clarify the statement "Although the disease was first reported in 1921 in Kenya, it has now spread to most SSA countries...". Has it spread or just been detected?
Resp: Clarified to read” The first documentation of ASF was in 1921 in Kenya and from there it has been reported in other countries inside and outside Africa”
we have also revised the sentence on line 86. "The last decade has seen a rapid increase in the pig production sector in Africa and a corresponding increase in detection of ASFV...".
Resp: The sentence has been revised as suggested by the reviewer.
Qn: What does the phrase on line 72 "..many ASFV genotypes have been established in SSA...". The review has not considered the evolution of ASFV. The sylvatic cycle is only mentioned on page 7 and should be included in the Introduction. The existence of a wild swine-tick cycle in SSA is critically important to both the evolution of ASFV and its emergence in domestic pigs.
Resp: Sentence corrected to read “All of the known 24 different ASFV genotypes have been established in SSA and an updated literature on the genotypes circulating in SSA is still needed”
Qn: Why is ASFV a 'large' DNA virus?
Resp: Classification of viruses looks beyond their diameters and genome sizes. A number of viruses including ASFV possess double stranded DNA as their genomes. These viruses are considered complex and have been classified as small or large depending on their type of replication. Large DNA viruses like ASFV encodes a virus-specific DNA polymerase responsible for their replication while small DNA viruses use a host DNA polymerase for their replication. Also, large DNA viruses encodes more proteins (eg. ASFV encodes for more than 150 proteins) than small viruses. A book chapter from K. V. Chaitanya, Genome and Genomics, https://doi.org/10.1007/978-981-15-0702-1_1 has information on this.
Qn: The sentence on line 112 needs revision. "Entry of the virus into pig cells (Fig2) is mostly via the tonsils closest to the lymph nodes" does not make sense. Infection by fomites could infect the animal via the tonsils but virus infects cells through a cell-surface receptor, implied in Figure 2.
Resp: We Agreed and the sentence is revised to read “ASFV entry into the cell is through the host’s cell surface receptors (Fig. 2) following an infection of the animal mostly via the tonsils closest to lymph nodes.
Qn: Line 115, virus in the bloodstream is normally referred to as viraemia. On line 119 ".. from one animal to another."
Resp: Line 115 is corrected to read “Following viraemia, the virus migrates to the tissue organs” Line 119 is also corrected.
Qn: on line 147, explain what pathogenicity is in the context of cell culture?
Resp: Corrected to read “Infectivity and cell response caused by the virus on cell cultures…”
Qn: When discussing the genome of ASFV, the authors should provide more background on the genome structure, nomenclature of ASFV genes and give more information on MGFs if they influence virulence.
The authors also do not consider the mutation rate of ASFV and whether this varies across the genome and how it has given rise to variation in key genes. This may be the only argument for complete genome sequencing.
Resp: The authors appreciate that ASF and ASFV had been reviewed by key experts in ASF and ASFV studies and the genome background, nomenclature as well as some information on virulence had been discussed. Such works have been cited but not discussed in details to avoid unnecessary repetitions.
Resp: The mutation rate of ASFV and whether it varies across the genome and how it has given rise to variation in key genes is a discussion which can be composed upon a comparison of multiple complete genomes from Sub-Saharan Africa, an area of focus for this review, but this is not possible with the few genomes currently available, including the only one full genotype II sequence released recently by Njau et al, 2021. Much as the authors agree that the information asked by the reviewer will form a good argument for complete genome sequencing, the complete genome sequences are needed before hand to be able to produce such an information.
Qn: The paragraph starting line 189 does not explain the transmission cycle, which is admittedly complex (warthog-tick-warthog; warthog-tick-domestic pig; warthog-domestic pig; domestic pig-domestic pig), particularly well. The natural cycle of wild pig - tick should be described and then the interaction between this and the introduction of domestic pigs. The distribution of key species such as warthogs in Africa is not considered but may contribute to the distribution of ASFV is SSA.
Resp: The authors agree that this review did not concentrate on the complexity of the disease transmission cycles and the distribution of ticks and warthogs in Africa. This was considered another area of focus which is broad enough for the writing of another review. The other aspects were briefly mentioned as the focus of the current review is in the ASFV Biology, Genomics and Genotypes Circulating in Sub-Saharan Africa.
Qn: Line 245: revise to "..on the genotyping of five ASFV isolates (or samples) from domestic..". Line 259. ".. where the transmission of the virus occurs.."
Resp: Sentences revised as suggested by the reviewer
Qn: The discussion on carrier status is mentioned on line 263 and again on 346. As this is a controversial subject, it only need one mention.
Resp: Agreed and amended.
Qn: On line 299, the discussion suggests that a delay in diagnosis is a major issue. Whilst accurate diagnosis of disease is important, some control measures can be implemented (movement restrictions / quarantine of infected properties) on suspicion of disease and can be effective, so whilst inconvenient, it is not an insurmountable problem.
Resp: The authors accept that disease suspicion has been used effectively and have added that in the paragraph. With this, we also can not overlook the fact that some diseases including ASF have symptoms highly related to other diseases and hence disease suspicion has sometimes created unnecessary panic to farmers making accurate disease diagnosis very important.
The discussion is now rephrased to read “Some control measures such as movement restrictions and quarantine of infected properties can and are being implemented on suspicion of disease. While disease suspicion has been implemented for a rescue, it is however notable that delays associated with remoteness hinder outbreak responses especially for diseases like ASF whose symptoms are closely related to other diseases.”
Qn: Line 392, why is genotyping using p72 not always sufficient for typing resolution or phenotype discrimination? What did further assessment using p54, p30 and B602L genes provide? Despite 12 pages of text it is not clear what the improved resolution gives for practical control of ASFV or in the understanding of its epidemiology/evolution. Based on the review it is surely possible to give a more nuanced recommendation than more sequencing across SSA. Should effort be directed at areas with persistent outbreaks or regions showing high genotype diversity.
Resp: The p72 region is important and useful for the diagnosis of ASF as it is a region that is well preserved in ASFV. It also has the hyper variable end that is used for genotyping the viruses into the 24 genotypes. The p54 and p30 are used for further discrimination as highlighted in the text. These regions though useful, they do not provide all of the information. For example, when the genotype II virus from Tanzania was compared with a genotype II virus from Georgia, the p72, p54 and p30 regions indicated that the virus strains were identical. Upon a comparison of their full genomes, it was observed that the two virus strains had a series of divergences in 5ʹ and 3ʹ multicopy gene families (MGFs). Apart from that, there is an insertion of 16 bases that yields a tandem repeat (aaaaaaataaacaaca) located at the 5ʹ of the ATG codon of locus KP177R that encodes for the K177R structural protein. Analysis of the numbers of reads confirmed the presence of this mutation which differentiated Tanzania/Rukwa/2017/1 from all currently available genotype II genome sequences. In addition to the multiple changes in the terminal multicopy gene families, the single-copy gene ASFV_Ch_ACD_00120 appeared to be absent from Tanzania/Rukwa/2017/1 but is present in Georgia 2007/1 and its derivatives. This is a practical example of what improved resolution can contribute on practical control of ASFV and in the understanding of its epidemiology/evolution. This information has been added in the manuscript.
With much thanks.
Reviewer 3 Report
Njau et al described a review manuscript “ASF: biology, genomics and genotypes circulating in Sub-Saharan Africa. Authors give overall good background of ASF in Sub-Saharan Africa. This is a valuable research article. This manuscript is suitable for publish in Viruses with minor revision.
Minor common:
- Combine the Figure 1 and Figure 3 together, since these two figs are quite similar.
Author Response
Dear Reviewer 3,
Thank you for your contributions towards the improvement of our manuscript.
I am seeking for a clarification on combining the images. Did you mean Figure 1 and Figure 2 or is it Plate 1 and Plate 2 that should be combined?
With much thanks,
Emma Njau
Round 2
Reviewer 2 Report
The revised manuscript has addressed the reviewers concerns and improved the manuscript.
Author Response
Dear Reviewer 2,
We are very grateful for your valuable contributions towards the improvement of our manuscript.
Thank you very much.